# Deterioration characteristics of cement-improved loess under dry–wet and freeze–thaw cycles

**Ying-jun Jiang[1], Chen-yang Ni[1]\*, Hong-wei Sha[2], Zong-hua Li[3], Lu-yao Cai[1]**

1 Key Laboratory for Special Area Highway Engineering of Ministry of Education, Chang' an University, Xi'an, China, 2 Shaanxi XiFa (North line) Intercity Railway Co., Ltd, Xi'an, China, 3 Shaanxi XiHan Intercity Railway Co., Ltd, Xi'an, China

\* chenyangni@chd.edu.cn

**Data Availability Statement:** All data and models generated or used during the study appear in the submitted article.

**Funding:** This work was supported by the Scientific Research of Central Colleges of China for Chang'an

## Abstract

The effects of cement dosage, compaction coefficient, molding method (vertical vibration method and static pressure method), and dry–wet and freeze–thaw cycles on the mechanical strength of cement-improved loess (CIL) were studied to reveal its strength degradation law under dry–wet and freeze–thaw cycles. Results show that when using the vertical vibration molding method, the strength degradation effect of CIL can be improved by increasing the cement dosage and compaction coefficient; however, it is not obvious. Under the action of dry–wet cycle, damages, such as voids and cracks of CIL, develop continuously. Further, the strength deteriorates continuously and does not decrease after more than 15 dry–wet cycles. Therefore, the dry–wet cycle degradation system is selected by considering the most unfavorable conditions. In the process of freeze–thaw alternation, the pores and fissures of CIL develop and evolve continuously and the strength deteriorates continuously under the joint influence of water and low temperature. The strength tends to become stable after more than 12 freeze–thaw cycles. According to the safety principle, the deterioration coefficient of the freeze–thaw cycles is 0.3.

## 1 Introduction

The main engineering problems of loess foundation are large pores, poor cementation, weak shear strength and large deformation under the action of external force. By adding an appropriate amount of ordinary Portland cement to treat loess, the strength of loess can be improved to meet the needs of foundation safety. The periodically changing climate, however, causes the cement-improved loess (CIL) subgrade to always alternate between saturated and unsaturated under the effect of rainfall and evaporation. Further, its physical and mechanical properties exhibit dynamic change under the continuous dry–wet cycles [1]. The subgrade soil is frozen in winter in the seasonally frozen soil area. The stress field of the subgrade soil changes and redistributes because of the joint action of temperature, water, and load, resulting in frost heaving, thawing settlement, frost boiling, and other frost damage phenomena. These phenomena cause many difficulties in railway engineering [2,3]. Some scholars have conducted

University (No. 300102218212); the scientific project from Shaanxi Provincial Communication No. 19-27K and the scientific project from Shaanxi Provincial Communication No. 18-02K. Shaanxi XiFa (North line) Intercity Railway Co., Ltd provided funding in the form of salary to Hong-wei Sha. Shaanxi XiHan Intercity Railway Co., Ltd provided funding in the form of salary to Zong-hua Li. The funders but did not have any role in the study design, data collection and analysis, decision to publish, or preparation of the manuscript. The specific roles of these authors are articulated in the 'author contributions' section.

**Competing interests:** Hong-wei Sha is a paid employee of commercial company Shaanxi XiFa (North line) Intercity Railway Co., Ltd. Zong-hua Li is a paid employee of commercial company Shaanxi XiHan Intercity Railway Co., Ltd. Shaanxi Provincial Communication provided funding in the form of grants for this study. There are no patents, products in development or marketed products to declare. This does not alter our adherence to PLOS ONE policies on sharing data and materials.

studies to understand the change law of the mechanical properties of loess under the action of dry–wet and freeze–thaw cycles.

First, engineering properties of collapsible loess stabilized by cement have been studied. Liu et al. [4] found that the cement improved microstructure of loess, increased clay content and enhanced cementation, thus increased the shear strength. Cui et al. [5] pointed out that the mineral composition of loess is changed by adding cement kiln dust, and a better packing and more stable soil structure are formed. Su [6] thought the strength of loess increases along with the mixture ratio increases, when the cement mixture ratio between 4%-14%, the scope of increases is obvious when the ages are certain.

Then, the dry-wet and freeze-thaw performance of remolded loess has been studied by researchers. For instance, Ni et al. [7] combined with the water holding capacity curve of loess, the microstructure evolution mechanism of two kinds of loess particles in the process of dry wet cycle was analyzed. The deterioration model of compacted loess considering the influence factors is established by Hu et al. [8]. Zhang et al. [9] considered that compared with unmodified loess, lime fly ash loess has higher strength and better water stability, which can be used as road subgrade in loess area. Bai et al. [10] hold that with the increase of dry wet cycles and the increase of cycle amplitude, the cohesion and internal friction angle of loess decrease, and the shear strength decreases significantly. Zhang et al. [11] found that freeze-thaw cycles resulted in increase in soil volume, decrease in moisture content, but negligible change in Atterberg limits. Wang et al. [12,13] studied volume change behavior and microstructure of stabilized loess under cyclic freeze-thaw conditions, and suggested the mix proportions of the three additives was recommended to be 4 to 5% cement, 6% lime, and 10% fly ash. Zhang et al. [14] thought that addition of cement to loess was effective to maintain zero mass loss and negligible frost susceptibility.

Loess is green building material according with contemporary call for sustainable development [15]. The above studies mainly focus on the influence of chemical improvement methods and cement content on the resistance of improved loess to dry–wet and freeze–thaw cycles. However, the strength of CIL is affected by internal and external factors under the action of the dry–wet and freeze–thaw cycles. The internal factors are mainly the compaction coefficient and cement dosage, whereas the external factors are mainly dry–wet and freeze–thaw cycles. Therefore, the vertical vibration compaction method (VVCM) and quasi-static compaction method (QSCM) are used to obtain CIL specimens. Further, the strength degradation characteristics of CIL under different cement dosages, compaction coefficients, and dry–wet and freeze–thaw cycles are studied to provide reference for engineering practice.

## 2 Materials and test methods

### 2.1 Materials

The loess used in this paper was obtained from Xi'an–Hancheng Intercity Railway. The physical properties of loess are presented in Table 1. The test cement is ordinary Portland cement P.

**Table 1. Physical properties of loess.**

| Particle density(g/cm³) | Liquid limit(%) | Plastic limit(%) | Plasticity index | Mass percentage corresponding to particle size (mm) | | | | |
|---|---|---|---|---|---|---|---|---|
| | | | | 0.25~0.75 | 0.05~0.25 | 0.01~0.05 | 0.005~0.01 | ≤0.005 |
| 2.74 | 26.4 | 15.7 | 10.7 | 2.47 | 7.22 | 53.43 | 13.83 | 23.05 |

**Table 2. Technical properties of cement.**

| Fineness (%) | Soundness | Loss on ignition (%) | Initial setting time (min) | Final setting time (min) |
|---|---|---|---|---|
| 1.2 | Qualified | 1.02 | 265 | 320 |

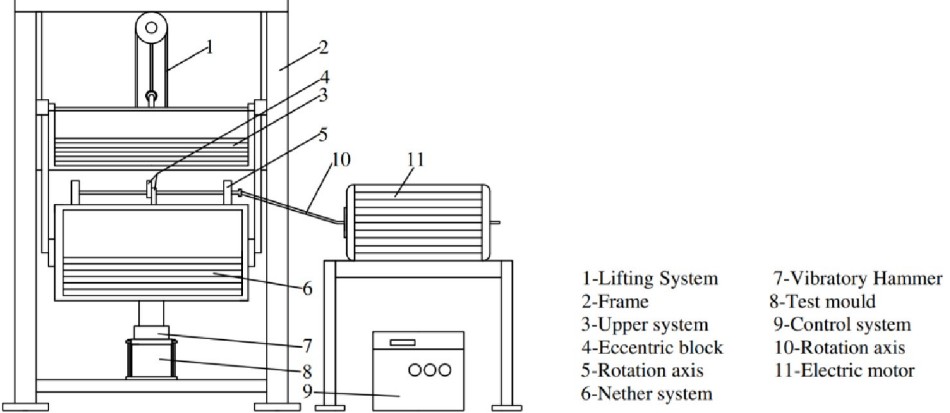

**Fig 1. Schematic of VVTE.**

O42.5 produced by Shaanxi Yaobai Special Cement Co., Ltd.; its technical properties are presented in Table 2. All the used materials satisfy the requirements of the Chinese specification of JTG/T F20-2015 [16].

## 2.2 Specimen preparation methods

(1) VVCM.   Cylindrical SCL specimens with a diameter and height of 100 mm were prepared as follows. First, the maximum dry density (MDD) and optimum water content (OWC) of the SCL specimens were determined via VVCM with 140 s of vertical vibration. Second, CSL was fabricated via VVCM according to the MDD and OWC. Finally, the prepared CSL was wrapped in plastic bags and stored in a standard curing room with a temperature of 20˚C ± 2˚C and a humidity of more than 95%.

VVTE was the core of the VVCM. Fig 1 illustrates the structure of the VVTE. The working parameters of the VVTE included a vibration frequency of 35 Hz, a nominal amplitude of 1.2 mm, and upper- and lower-system weights of 120 and 180 kg, respectively [17–21].

(2) QSCM.   The CIL is pressed to the specified height according to the static pressure method in the code for geotechnical test of railway engineering (TB10102-2010) (hereinafter referred to as the "code") [22].

## 2.3 Test plan and methods

### 2.3.1 Test plan.

1. The influence of cement dosage on the strength deterioration of improved loess was studied under the action of the dry–wet and freeze–thaw cycles. The cement dosage (denoted as $P_s$) is proposed to be 2%, 3%, 4%, and 6%.

2. The influence of compaction coefficient on the strength deterioration of improved loess was studied under the action of the dry–wet and freeze–thaw cycles. The coefficient of compaction (denoted as $K$) is proposed to be 0.92, 0.95, and 0.97.

3. The influence of the forming method on the strength deterioration of the improved loess was studied under the action of the dry–wet and freeze–thaw cycles. The forming method is planned to adopt the VVCM and QSCM.

**2.3.2 Test methods.** *(1) Dry–wet cycle test.* The specimen was taken out of the curing room on the final day of the test-piece curing period. Its mass $m_0$ was weighed before it was soaked in water at 20˚C ± 2˚C for one day and night to achieve saturation. After soaking, the surface moisture was wiped off using a wet cloth. Then, the mass $m_1$ under the saturated state was determined, and the dry–wet cycle test was performed. In the drying and wetting cycles, the saturated specimens were placed in a cool and ventilated place for natural air drying. The mass $m$ was weighed after every 2 h during the air drying process. At $m = (m_0 ± 5)$ g, the specimen can be considered to have reached the optimum moisture content, and air drying is stopped. After the specimen was air dried to the optimal moisture content, it was immersed in water maintained at 20˚C ± 2˚C for 16 h (i.e., one dry–wet cycle).

The unconfined compressive strength test was conducted when the specimen was subjected to the corresponding dry–wet cycle for the specified number of rimes, and the loading rate was 1 mm/min. The dry-to-wet residual strength ratio was calculated according to formula (1) and used to characterize the compressive strength stability of the specimens under dry–wet cycles.

$$\eta_G = \frac{q_{Gu}}{q_u} \times 100, \tag{1}$$

where $\eta_G$ is the dry–wet residual strength ratio (%), $q_{Gu}$ is the compressive strength of the specimens after n dry–wet cycles (MPa), and $q_u$ is the unconfined compressive strength of the specimens (MPa).

*(2) Freeze–thaw cycle test.* The specimens were soaked in water maintained at 20˚C until the final day of curing. After soaking, the surface water was wiped off using a wet cloth to complete the freeze–thaw cycle. The freezing temperature was −18˚C, and the freezing time was 16 h. Water thawing was the melting method, the melting temperature was 20˚C ±2˚C, and the melting time was 8 h (the time required for one freeze–thaw cycle).

Once the specimen was subjected to the freeze–thaw cycle for the specified number of times, the unconfined compressive strength test was conducted, and the loading rate was considered to be 1 mm/min. The ratio of freeze–thaw residual strength of the improved loess can be calculated according to formula (2). This ratio represents the stability of compressive strength under freeze–thaw cycles.

$$\eta_D = \frac{q_{Du}}{q_u} \times 100, \tag{2}$$

where $\eta_D$ is the residual strength ratio after freeze–thaw cycles (%) and $q_{Du}$ is the compressive strength of the specimens after n freeze–thaw cycles (MPa).

## 3 Results and analysis

### 3.1 Dry–wet cycle test

**3.1.1 Test result.** The test results of CIL under dry–wet cycles are presented in Table 3.

It can be seen from Table 3 that under the same other objective conditions, with the increase of cement dosage and compaction coefficient, the compressive strength of cement modified loess increases gradually after drying and wetting cycles; With the increase of dry wet cycles, the compressive strength of cement improved loess decreases gradually, and tends to be stable after a certain number of dry–wet cycles.

**3.1.2 Analysis of the influencing factors.** There is assumed to be a strength deterioration coefficient equation for CIL under the condition of dry–wet cycle. The following three

**Table 3. 28d compressive strength of CIL under dry–wet cycles.**

| Forming method | $P_S$ (%) | K | Compressive strength of CIL under the following cycles (MPa) | | | | | | | | | |
|---|---|---|---|---|---|---|---|---|---|---|---|---|
| | | | 0 | 1 | 3 | 5 | 7 | 9 | 12 | 15 | 20 | 25 |
| VVCM | 2 | 0.92 | 1.13 | 0.83 | 0.72 | 0.64 | 0.63 | 0.54 | 0.49 | 0.48 | 0.48 | 0.47 |
| | | 0.95 | 1.53 | 1.22 | 1.05 | 0.94 | 0.84 | 0.81 | 0.71 | 0.67 | 0.67 | 0.66 |
| | | 0.97 | 2.00 | 1.63 | 1.38 | 1.27 | 1.20 | 1.03 | 0.98 | 0.94 | 0.94 | 0.92 |
| | 3 | 0.92 | 1.47 | 1.17 | 1.01 | 0.93 | 0.80 | 0.74 | 0.65 | 0.64 | 0.63 | 0.61 |
| | | 0.95 | 1.82 | 1.48 | 1.35 | 1.18 | 1.04 | 0.98 | 0.91 | 0.85 | 0.80 | 0.77 |
| | | 0.97 | 2.46 | 2.06 | 1.83 | 1.69 | 1.38 | 1.25 | 1.19 | 1.13 | 1.11 | 1.11 |
| | 4 | 0.92 | 1.62 | 1.33 | 1.18 | 1.06 | 0.86 | 0.82 | 0.76 | 0.73 | 0.71 | 0.68 |
| | | 0.95 | 2.22 | 1.89 | 1.69 | 1.57 | 1.29 | 1.20 | 1.08 | 1.02 | 0.98 | 0.92 |
| | | 0.97 | 2.74 | 2.44 | 2.18 | 2.07 | 1.59 | 1.45 | 1.35 | 1.26 | 1.26 | 1.25 |
| | 6 | 0.92 | 1.95 | 1.67 | 1.50 | 1.33 | 1.20 | 1.12 | 1.10 | 1.05 | 1.03 | 1.87 |
| | | 0.95 | 2.49 | 2.13 | 2.00 | 1.74 | 1.56 | 1.46 | 1.29 | 1.28 | 1.26 | 2.44 |
| | | 0.97 | 3.02 | 2.60 | 2.51 | 2.06 | 1.90 | 1.80 | 1.67 | 1.65 | 1.64 | 2.95 |
| QSCM | 2 | 0.92 | 1.03 | 0.81 | 0.69 | 0.59 | 0.54 | 0.50 | 0.46 | 0.45 | 0.43 | 0.42 |
| | | 0.95 | 1.34 | 1.07 | 0.92 | 0.80 | 0.72 | 0.66 | 0.60 | 0.58 | 0.56 | 0.55 |
| | | 0.97 | 1.65 | 1.32 | 1.16 | 1.01 | 0.89 | 0.83 | 0.74 | 0.73 | 0.71 | 0.69 |
| | 3 | 0.92 | 1.30 | 1.04 | 0.88 | 0.75 | 0.69 | 0.64 | 0.59 | 0.57 | 0.55 | 0.53 |
| | | 0.95 | 1.60 | 1.30 | 1.10 | 0.99 | 0.86 | 0.82 | 0.74 | 0.70 | 0.69 | 0.67 |
| | | 0.97 | 2.03 | 1.68 | 1.44 | 1.26 | 1.10 | 1.02 | 0.93 | 0.89 | 0.85 | 0.85 |
| | 4 | 0.92 | 1.49 | 1.22 | 1.04 | 0.89 | 0.83 | 0.76 | 0.72 | 0.66 | 0.64 | 0.63 |
| | | 0.95 | 1.90 | 1.58 | 1.35 | 1.18 | 1.06 | 0.99 | 0.91 | 0.84 | 0.82 | 0.80 |
| | | 0.97 | 2.26 | 1.90 | 1.63 | 1.42 | 1.29 | 1.18 | 1.08 | 1.02 | 0.97 | 0.97 |
| | 6 | 0.92 | 2.03 | 1.68 | 1.42 | 1.24 | 1.14 | 1.04 | 0.95 | 0.89 | 0.87 | 0.85 |
| | | 0.95 | 2.44 | 2.03 | 1.76 | 1.51 | 1.39 | 1.27 | 1.17 | 1.07 | 1.05 | 1.02 |
| | | 0.97 | 2.80 | 2.38 | 2.02 | 1.76 | 1.62 | 1.48 | 1.34 | 1.26 | 1.23 | 1.20 |

boundary conditions are satisfied.

$$\text{When } N = 0, \ \eta_{GN} = \eta_{G0},$$

$$\text{when } N = \infty, \ \eta_{GN} = \eta_{G\infty},$$

$$\eta_{G0} > \eta_{G\infty},$$

where $N$ is the number of dry–wet cycles of CIL, $\eta_{GN}$ is the strength degradation coefficient of CIL after $N$ dry–wet cycles, $\eta_{G0}$ is the strength degradation coefficient of CIL without dry–wet cycles, i.e., 1, and $\eta_{G\infty}$ is the ultimate dry–wet strength degradation coefficient of CIL.

According to the above boundary conditions, the strength deterioration coefficient equation of CIL after dry–wet cycle is established as

$$\eta_{GN} = \eta_{G\infty} - \frac{\eta_{G\infty} - 1}{\xi \cdot N^2 + 1}, \tag{3}$$

where $\xi$ is the regression parameter to be determined.

The strength degradation equation of CIL after the dry–wet cycles is obtained by fitting with formula (3), as shown in Fig 2. The deterioration coefficient of CIL can be calculated according to formula (1).

The following observations are obtained from Fig 2.

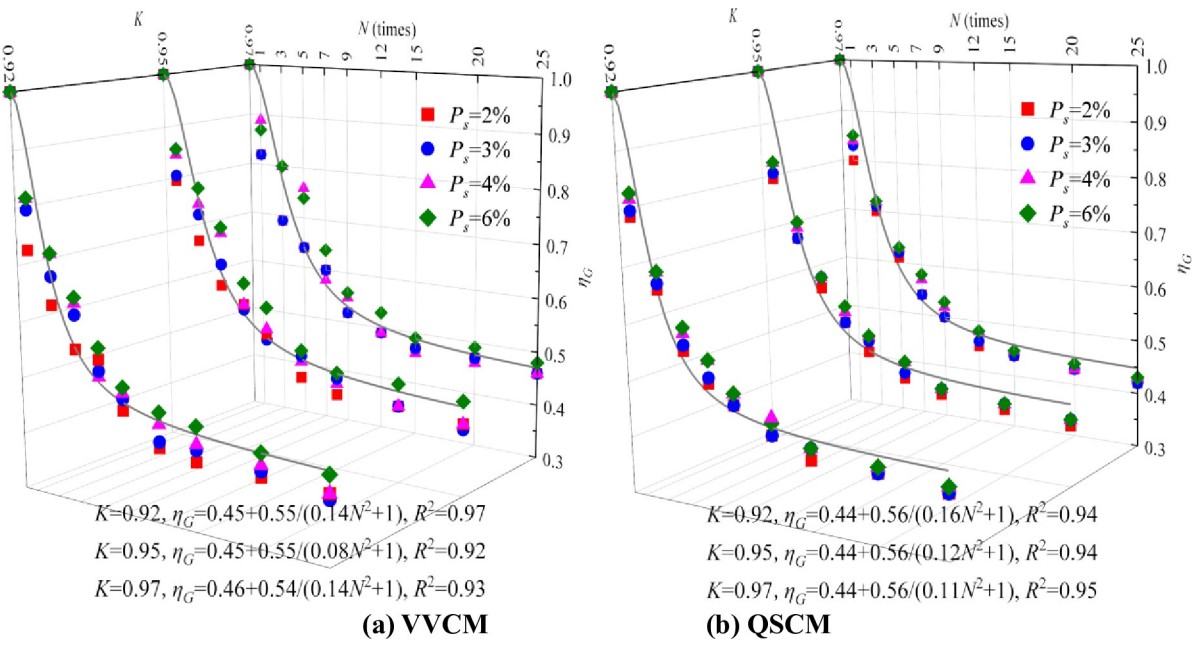

$K=0.92, \eta_G=0.45+0.55/(0.14N^2+1), R^2=0.97$
$K=0.95, \eta_G=0.45+0.55/(0.08N^2+1), R^2=0.92$
$K=0.97, \eta_G=0.46+0.54/(0.14N^2+1), R^2=0.93$

$K=0.92, \eta_G=0.44+0.56/(0.16N^2+1), R^2=0.94$
$K=0.95, \eta_G=0.44+0.56/(0.12N^2+1), R^2=0.94$
$K=0.97, \eta_G=0.44+0.56/(0.11N^2+1), R^2=0.95$

**(a) VVCM**       **(b) QSCM**

**Fig 2. Strength degradation equation of CIL with different $K$ after dry–wet cycles.**

1. With the increasing $P_s$ and $K$, the deterioration coefficient of CIL increases even though it is not obvious. The influence of $P_s$ and $K$ on the degradation coefficient becomes less obvious with the increasing number of dry–wet cycles.

2. Compared with QSCM, the deterioration coefficient of CIL formed via VVCM is higher even though it is not obvious. The effect of the molding method on the degradation coefficient becomes less obvious with the increasing number of dry–wet cycles.

3. The deterioration coefficient of CIL gradually decreases and the decline rate gradually decreases with the increasing number of dry–wet cycles. The deterioration coefficient is stable at 0.41–0.48 when the number of dry–wet cycles exceeds 15. The degradation coefficient of a dry–wet cycle is 0.40 when considering the most unfavorable conditions. The larger the coefficient $\xi$, the smaller will be the strength deterioration coefficient of CIL after $N$ dry–wet cycles.

**3.1.3 Mechanism analysis.** Results show that the strength of CIL tends to deteriorate under the action of dry–wet cycles, which can be attributed to the irreversible transformation of its microstructure.

1. A certain amount of soluble salt is present in loess, which will result in the formation of salt cement on the surface of soil particles, providing support for the CIL structure. The strength of CIL can be mainly attributed to the skeleton structure formed via multiparty cementation after cement hydration [23].

2. A large amount of soluble salt dissolves in the improved soil during the humidification stage, which leads to the increase and enlargement of the pores in CIL. Further, reinforcement cohesion that can be attributed to the cementation of soluble salt can be observed, which will damage the skeleton structure formed by cement.

3. In the air drying stage, CIL is subjected to the drying shrinkage process. The drying shrinkage process will promote the appearance and development of cracks on the surface and inside of the improved loess. Simultaneously, water evaporation will cause the migration and precipitation of soluble salts, resulting in the dispersion of clay particles in CIL [24]. Although the promotion of cement hydration in the humidification stage, to some extent, will restore the strength of CIL, the recovery capacity is limited, so the overall performance of the compressive strength of the cement modified loess is reduced.

## 3.2 Freeze–thaw cycle test

**3.2.1 Test result.** The test results related to the compressive strength of CIL under freeze–thaw cycles are presented in Table 4. After ten freeze–thaw cycles, CIL specimen with $P_s$ = 2% was loose; therefore, no data were recorded.

It can be seen from Table 4 that under the same objective conditions, with the increase of cement dosage and compaction coefficient, the compressive strength of cement modified loess after freeze-thaw cycle increases gradually; With the increase of freeze-thaw cycles, the compressive strength of cement modified loess decreases gradually, and tends to be stable after a certain number of freeze-thaw cycles.

**3.2.2 Analysis of the influencing factors.** There is assumed to be a strength deterioration coefficient equation for CIL under freeze–thaw cycles, which satisfies the following three

**Table 4. 28d compressive strength of CIL under freeze–thaw cycles.**

| Forming method | K | $P_s$ (%) | Compressive strength of CIL under the following cycles (MPa) | | | | | | | | |
|---|---|---|---|---|---|---|---|---|---|---|---|
| | | | 0 | 1 | 3 | 5 | 7 | 9 | 12 | 15 | 20 |
| VVCM | 0.92 | 2 | 1.13 | 0.81 | 0.70 | 0.53 | 0.35 | 0.27 | - | - | - |
| | | 3 | 1.47 | 1.09 | 0.93 | 0.73 | 0.56 | 0.47 | 0.44 | 0.41 | 0.39 |
| | | 4 | 1.62 | 1.25 | 1.09 | 0.84 | 0.71 | 0.65 | 0.59 | 0.57 | 0.56 |
| | | 6 | 2.29 | 1.76 | 1.54 | 1.19 | 1.01 | 0.9 | 0.85 | 0.81 | 0.84 |
| | 0.95 | 2 | 1.53 | 1.14 | 0.96 | 0.73 | 0.53 | 0.36 | - | - | - |
| | | 3 | 1.82 | 1.39 | 1.24 | 0.92 | 0.82 | 0.77 | 0.7 | 0.68 | 0.65 |
| | | 4 | 2.22 | 1.77 | 1.56 | 1.26 | 1.04 | 0.95 | 0.88 | 0.91 | 0.87 |
| | | 6 | 2.68 | 2.29 | 1.96 | 1.6 | 1.43 | 1.27 | 1.22 | 1.14 | 1.13 |
| | 0.97 | 2 | 2.00 | 1.53 | 1.26 | 0.99 | 0.70 | 0.50 | - | - | - |
| | | 3 | 2.46 | 1.94 | 1.68 | 1.34 | 1.19 | 1.03 | 1.02 | 0.99 | 0.92 |
| | | 4 | 2.74 | 2.30 | 2.01 | 1.68 | 1.38 | 1.29 | 1.20 | 1.14 | 1.15 |
| | | 6 | 3.39 | 2.77 | 2.39 | 2.03 | 1.78 | 1.5 | 1.46 | 1.48 | 1.37 |
| QSCM | 0.92 | 2 | 1.03 | 0.71 | 0.61 | 0.46 | 0.31 | 0.24 | - | - | - |
| | | 3 | 1.30 | 0.95 | 0.81 | 0.64 | 0.49 | 0.42 | 0.39 | 0.37 | 0.35 |
| | | 4 | 1.49 | 1.13 | 0.98 | 0.76 | 0.64 | 0.58 | 0.53 | 0.52 | 0.52 |
| | | 6 | 2.03 | 1.56 | 1.37 | 1.05 | 0.91 | 0.83 | 0.76 | 0.74 | 0.73 |
| | 0.95 | 2 | 1.34 | 0.96 | 0.81 | 0.62 | 0.43 | 0.29 | | – | - |
| | | 3 | 1.60 | 1.19 | 1.04 | 0.81 | 0.71 | 0.64 | 0.6 | 0.58 | 0.57 |
| | | 4 | 1.90 | 1.44 | 1.28 | 1.02 | 0.9 | 0.83 | 0.77 | 0.75 | 0.74 |
| | | 6 | 2.44 | 1.89 | 1.61 | 1.32 | 1.16 | 1.05 | 0.99 | 0.98 | 0.96 |
| | 0.97 | 2 | 1.65 | 1.22 | 1.01 | 0.80 | 0.57 | 0.39 | - | - | - |
| | | 3 | 2.03 | 1.53 | 1.32 | 1.05 | 0.93 | 0.84 | 0.78 | 0.75 | 0.73 |
| | | 4 | 2.26 | 1.76 | 1.55 | 1.30 | 1.11 | 1.02 | 0.96 | 0.93 | 0.93 |
| | | 6 | 2.80 | 2.18 | 1.87 | 1.59 | 1.38 | 1.23 | 1.16 | 1.14 | 1.11 |

boundary conditions:

$$\text{when } N = 0, \ \eta_{\text{DN}} = \eta_{\text{D0}},$$

$$\text{when } N = \infty, \ \eta_{\text{DN}} = \eta_{\text{D}\infty},$$

$$\eta_{\text{D0}} > \eta_{\text{D}\infty},$$

where $N$ is the number of freeze–thaw cycles of CIL, $\eta_{\text{DN}}$ is the strength degradation coefficient of CIL after $N$ freeze–thaw cycles, $\eta_{\text{D0}}$ is the strength degradation coefficient of CIL without freeze–thaw cycles, i.e., 1, and $\eta_{\text{D}\infty}$ is the ultimate freeze–thaw strength degradation coefficient of CIL.

According to the above boundary conditions, the strength deterioration coefficient equation of CIL after freeze–thaw cycle can be established.

$$\eta_{DN} = \eta_{D\infty} - \frac{\eta_{D\infty} - 1}{\xi \cdot N^2 + 1}, \tag{4}$$

where $\xi$ is the regression parameter to be determined.

The freeze–thaw deterioration coefficient equation of CIL can be obtained by Eq (4), as shown in Fig 3. The deterioration coefficient of CIL after the freeze–thaw cycle can be calculated using Eq (2). The data of CIL with $P_s = 2\%$ is missing; therefore, fitting processing is not required.

The following can be observed from Fig 3.

1. With the increasing $P_s$, the deterioration coefficient of the freeze–thaw strength of CIL increases, especially when $P_s$ increases from 2% to 3%. The deterioration coefficient of CIL can be increased by at least 60% after nine freeze–thaw cycles. When $P_s$ is greater than or

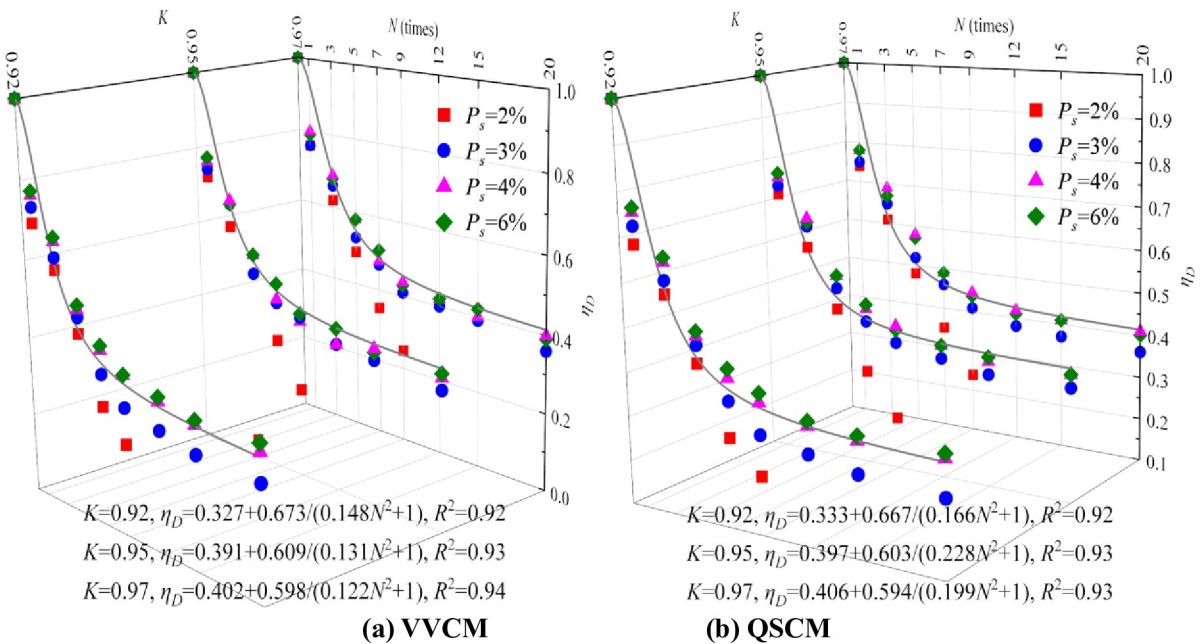

**Fig 3. Strength degradation equation of CIL with different $K$ after freeze–thaw cycles.**

equal to 3%, the influence of Ps on the deterioration coefficient of the freeze–thaw strength of CIL can be ignored.

2. With the increasing $K$, the deterioration coefficient of CIL increases even though it is not obvious. The effect of $K$ on the deterioration coefficient becomes less obvious with the increasing number of freeze–thaw cycles. When $K$ is greater than or equal to 0.95, the increasing value of $K$ has little effect on improving the freeze–thaw resistance of CIL.

3. Compared with QSCM, the deterioration coefficient of CIL formed by VVCM is increased even though it is not obvious. The effect of molding method on the degradation coefficient becomes less obvious with the increasing number of freeze–thaw cycles.

4. With the increasing number of freeze–thaw cycles, the deterioration coefficient of CIL gradually decreases and the decline rate gradually decreases. Based on Table 3, it can be seen that when the number of freeze–thaw cycles tends to $\infty$, the deterioration coefficients of CIL of the two molding methods are stable at 0.33–0.40. The degradation coefficient of the freeze–thaw cycle is 0.3 when considering the most unfavorable conditions.

**3.2.3 Mechanism analysis.** When the soil sample does not experience freeze–thaw cycle, the soil particles are arranged compactly. Further, the main structural types are flocculation and aggregates. Majority of the soil particles are aggregates and agglomerates, wherein many small particles are attached to large particles, and most of the particles are in face-to-face contact. With the increasing number of freeze–thaw cycles, the internal structure of soil becomes unbalanced under the joint influence of water and low temperature and the frost heaving damage is intensified, resulting in a continuous increase in the cement improvement strength.

1. When the soil is frozen, the free water in the soil pores condenses into ice to form a cold structure. The expansion force produced squeezes the surrounding particles because the volume of ice is larger than that of water. The large particle system gradually disintegrates under the effect of frost heaving force, the particles gradually disintegrate, and the pores increase.

2. When the soil is thawed, the ice crystals filled in the pores melt into water and gradually dissipate. However, the soil particle skeleton cannot completely return to the initial state, resulting in the continuous change of the pore morphology and the continuous development and evolution of pores and fissures in the soil, indicating that soil pores increase with the increasing number of freeze–thaw cycles [25].

3. In case of the freeze–thaw cycle, with the migration of water, soil pores, as the natural channels of water migration, are constantly scoured and part of the clay minerals are dissolved, resulting in increasing soil pores. At this time, the contact modes of the particles are mainly point-to-face and point-to-point. However, after repeated freeze–thaw cycles, the influence of soil failure gradually decreases, the internal structure of soil reaches a new balance, and the strength deterioration coefficient reaches an equilibrium value.

## 4 Conclusion

1. When using VVCM, increasing the cement dosage and compaction coefficient can improve the strength degradation effect of CIL even though it is not obvious.

2. Under the action of dry–wet cycle, damage, such as voids and cracks, develop continuously in CIL, and the strength deteriorates continuously. The strength does not decrease after

more than 15 dry–wet cycles. The deterioration coefficient of the dry–wet cycle is 0.40 when considering the most unfavorable conditions.

3. In the process of freeze–thaw alternation, under the joint influence of water and low temperature, the pores and fissures of CIL develop and evolve continuously and the strength deteriorates continuously. The strength tends to become stable after more than 12 freeze–thaw cycles. According to the safety principle, the deterioration coefficient of the freeze–thaw cycles is 0.3.

## Author Contributions

**Data curation:** Chen-yang Ni.

**Formal analysis:** Ying-jun Jiang, Chen-yang Ni.

**Funding acquisition:** Ying-jun Jiang.

**Investigation:** Hong-wei Sha, Zong-hua Li, Lu-yao Cai.

**Software:** Chen-yang Ni.

**Writing – original draft:** Chen-yang Ni.

**Writing – review & editing:** Chen-yang Ni, Lu-yao Cai.

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
