## [Decision Letter · Decision Letter 0]

29 Apr 2021

PONE-D-21-11948

Deterioration Characteristics of Cement-Improved Loess under Dry–wet and Freeze–thaw Cycles

PLOS ONE

Dear Dr. Ni,

Thank you for submitting your manuscript to PLOS ONE. After careful consideration, we feel that it has merit but does not fully meet PLOS ONE’s publication criteria as it currently stands. Therefore, we invite you to submit a revised version of the manuscript that addresses the points raised during the review process.

Please consider all the comments of all reviewers including reviewer 3.

We look forward to receiving your revised manuscript.

Kind regards,

Ahmed Mancy Mosa, Ph.D.

Academic Editor

PLOS ONE

Journal Requirements:

PLOS requires an ORCID iD for the corresponding author in Editorial Manager on papers submitted after December 6th, 2016. Please ensure that you have an ORCID iD and that it is validated in Editorial Manager. To do this, go to ‘Update my Information’ (in the upper left-hand corner of the main menu), and click on the Fetch/Validate link next to the ORCID field. This will take you to the ORCID site and allow you to create a new iD or authenticate a pre-existing iD in Editorial Manager. Please see the following video for instructions on linking an ORCID iD to your Editorial Manager account: https://www.youtube.com/watch?v=_xcclfuvtxQ

The authors declare no conﬂicts of interest.

We note that one or more of the authors are employed by a commercial company: Shaanxi XiHan Intercity Railway Co., Ltd and Shaanxi XiFa (North line) Intercity Railway Co., Ltd

Reviewers' comments:

Reviewer's Responses to Questions

**Comments to the Author**

1. Is the manuscript technically sound, and do the data support the conclusions?

Reviewer #1: Yes

Reviewer #2: Yes

Reviewer #3: No

2. Has the statistical analysis been performed appropriately and rigorously? 

Reviewer #1: N/A

Reviewer #2: Yes

Reviewer #3: No

3. Have the authors made all data underlying the findings in their manuscript fully available?

Reviewer #1: Yes

Reviewer #2: Yes

Reviewer #3: Yes

4. Is the manuscript presented in an intelligible fashion and written in standard English?

Reviewer #1: Yes

Reviewer #2: Yes

Reviewer #3: No

5. Review Comments to the Author

Reviewer #1: In this paper, the strength degradation of Cement Improved Loess formed by vertical vibration method under dry-wet and freeze-thaw cycles is studied, and the degradation equation and degradation coefficient are proposed. The overall structure is complete, with good logic and detailed test data. But there are several problems：

1) Some sentences need to be considered, such as vertical vibration compaction method (VVCM) molding cement improved loess, the reliability of the method is evaluated, what method is evaluated?

2) The research status of Cement Improved Loess at home and abroad needs to be further understood and mastered.

3) Describe about the experimental case study in more detail.

4) The graphic information of vertical vibration compactor has been reflected in the corresponding references. Is it necessary to add it?

5) Some English abbreviations appearing for the first time in this paper need to be described in detail.

6) Part of the language needs to be further refined.

7) The data points in Fig. 2 and Fig. 3 should be further analyzed and explained.

8) The format of the paper should be arranged according to the requirements of the journal.

9) The analysis of the results and the discussion part is not deep enough and needs further analysis.

Reviewer #2: This paper studies the strength degradation of Cement Improved Loess formed by vertical vibration method under dry-wet and freeze-thaw cycles, which has good logic and detailed test data. The research topic has a certain engineering application value. Suggestions:

1) The research status of Cement Improved Loess at home and abroad needs to be further understood and mastered.

2) The fitting equations in Fig. 2 and Fig. 3 are best presented in tabular form.

3) The meaning of R2 in Fig. 2 and Fig. 3 needs to be explained.

4) The quality of English needs improving.

5) Language and writing should be concise.

6) The format shall meet the requirements of periodical format.

7) To facilitate readers to understand the test process, please add test photos of VVCM and QSCM.

8) Please add some results analysis for table 3 and table 4.

Reviewer #3: In this manuscript, based on dry–wet cycle and freeze–thaw cycle tests, the deterioration characteristics of mechanical strength for cement-improved loess has been studied. This work is interesting. However, the studies are very poor and low. Moreover, there are many serious defects in this paper.

The main comments are as follows,

1. The English is very poor and there are many language errors. I think it should be edited by one native speaker before it is submitted.

2. The section 1 is very bad and should be rewritten. The previous studies should be analyzed more comprehensively and deeply. And the objective of this paper should be provided clearly.

3. Line 89, what is SCL?

4. Line 91, what is CSL?

5. Line 95, what is VVTE?

6. Line 98, I do not know why those References have been provided here.

7. Lines 83 and 102, I think the specifications can be provided as the References.

8. Line 127, "for the specified number of rimes". is it right?

9. Line 144, there is a wrong. "Error! Reference source not found.."

10. In subsection 3.1.1, I think the test results should be described clearly before the table 3.

11. Lines 223-224, "The deterioration coefficient of CIL after the freeze–thaw cycle can be calculated using equation (2)." is it right?

12. Line 240, "Based on Table 3,", is it right?

13. The studies in section 3 are too low and simple. I think deep analysis should be conducted and the test results should be verified.

14. There are only four international papers in the Reference list. For one international manuscript, I think more international papers should be included in the Reference list.

6. PLOS authors have the option to publish the peer review history of their article (what does this mean?). If published, this will include your full peer review and any attached files.

Reviewer #1: No

Reviewer #2: No

Reviewer #3: No

---

## [Author Response · Author response to Decision Letter 0]

18 May 2021

Reviewer #1: In this paper, the strength degradation of Cement Improved Loess formed by vertical vibration method under dry-wet and freeze-thaw cycles is studied, and the degradation equation and degradation coefficient are proposed. The overall structure is complete, with good logic and detailed test data. But there are several problems：

1) Some sentences need to be considered, such as vertical vibration compaction method (VVCM) molding cement improved loess, the reliability of the method is evaluated, what method is evaluated?

To solve this problem, the words used in this paper are adjusted and modified, and the reliability of the VVCM is evaluated.

2) The research status of Cement Improved Loess at home and abroad needs to be further understood and mastered.

Introduction of this paper has been rewritten and more international papers are added.

3) Describe about the experimental case study in more detail.

Modified.

4) The graphic information of vertical vibration compactor has been reflected in the corresponding references. Is it necessary to add it?

For the sake of the integrity of the paper, the idea of my collaborators and I is that we need to add.

5) Some English abbreviations appearing for the first time in this paper need to be described in detail.

Modified.

6) Part of the language needs to be further refined.

Modified.

7) The data points in Fig. 2 and Fig. 3 should be further analyzed and explained.

After data processing, the graph is analyzed and explained in depth, see 3.1 and 3.2.

8) The format of the paper should be arranged according to the requirements of the journal.

Modified.

9) The analysis of the results and the discussion part is not deep enough and needs further analysis.

After data processing, the graph is analyzed and explained in depth, see 3.1 and 3.2.

Reviewer #2: This paper studies the strength degradation of Cement Improved Loess formed by vertical vibration method under dry-wet and freeze-thaw cycles, which has good logic and detailed test data. The research topic has a certain engineering application value. Suggestions:

1) The research status of Cement Improved Loess at home and abroad needs to be further understood and mastered.

Modified. The section 1 has been rewritten.

2) The fitting equations in Fig. 2 and Fig. 3 are best presented in tabular form.

My collaborators and I think that the fitting formula is more intuitive when it is placed on the edge of the curve.

3) The meaning of R2 in Fig. 2 and Fig. 3 needs to be explained.

Modified. Where R2 is the coefficient of determination. The R2 is between 0 and 1, and the closer to 1, the better the regression fitting effect.

4) The quality of English needs improving.

Modified.

5) Language and writing should be concise.

Modified.

6) The format shall meet the requirements of periodical format.

Modified.

7) To facilitate readers to understand the test process, please add test photos of VVCM and QSCM.

As the experiment was finished, no photos were taken at that time.

8) Please add some results analysis for table 3 and table 4.

Modified.

Reviewer #3: In this manuscript, based on dry–wet cycle and freeze–thaw cycle tests, the deterioration characteristics of mechanical strength for cement-improved loess has been studied. This work is interesting. However, the studies are very poor and low. Moreover, there are many serious defects in this paper.

The main comments are as follows,

1. The English is very poor and there are many language errors. I think it should be edited by one native speaker before it is submitted.

Modified.

2. The section 1 is very bad and should be rewritten. The previous studies should be analyzed more comprehensively and deeply. And the objective of this paper should be provided clearly.

Modified. The section 1 has been rewritten.

3. Line 89, what is SCL?

Mistakes in writing. Modified.

4. Line 91, what is CSL?

Mistakes in writing. Modified.

5. Line 95, what is VVTE?

Modified. VVTE is the abbreviation of vertical vibration testing equipment.

6. Line 98, I do not know why those References have been provided here.

The value of VVTE parameter in this paper is directly used according to the research results of the research group, so we need to quote these articles.

7. Lines 83 and 102, I think the specifications can be provided as the References.

Modified.

8. Line 127, "for the specified number of rimes". is it right?

Modified.

9. Line 144, there is a wrong. "Error! Reference source not found.."

This problem caused by the wrong use of cross references in writing have been corrected.

10. In subsection 3.1.1, I think the test results should be described clearly before the table 3.

Analysis of test results has been added in subsection 3.1.1.

11. Lines 223-224, "The deterioration coefficient of CIL after the freeze–thaw cycle can be calculated using equation (2)." is it right?

Yes, it’s right.

12. Line 240, "Based on Table 3,", is it right?

Modified.

13. The studies in section 3 are too low and simple. I think deep analysis should be conducted and the test results should be verified.

Modified.

14. There are only four international papers in the Reference list. For one international manuscript, I think more international papers should be included in the Reference list.

International papers have been added.

---

## [Decision Letter · Decision Letter 1]

31 May 2021

Deterioration Characteristics of Cement-Improved Loess under Dry–wet and Freeze–thaw Cycles

PONE-D-21-11948R1

Dear Dr. Ni,

We’re pleased to inform you that your manuscript has been judged scientifically suitable for publication and will be formally accepted for publication once it meets all outstanding technical requirements.

Kind regards,

Ahmed Mancy Mosa, Ph.D.

Academic Editor

PLOS ONE

Additional Editor Comments (optional):

Reviewers' comments:

Reviewer's Responses to Questions

**Comments to the Author**

1. If the authors have adequately addressed your comments raised in a previous round of review and you feel that this manuscript is now acceptable for publication, you may indicate that here to bypass the “Comments to the Author” section, enter your conflict of interest statement in the “Confidential to Editor” section, and submit your "Accept" recommendation.

Reviewer #1: All comments have been addressed

Reviewer #2: All comments have been addressed

2. Is the manuscript technically sound, and do the data support the conclusions?

Reviewer #1: Yes

Reviewer #2: Yes

3. Has the statistical analysis been performed appropriately and rigorously? 

Reviewer #1: Yes

Reviewer #2: Yes

4. Have the authors made all data underlying the findings in their manuscript fully available?

Reviewer #1: Yes

Reviewer #2: Yes

5. Is the manuscript presented in an intelligible fashion and written in standard English?

Reviewer #1: Yes

Reviewer #2: Yes

6. Review Comments to the Author

Reviewer #1: The issues raised have been resolved and corrected. The article describes the strength degradation of cement-improved loess under dry, wet and freeze-thaw conditions. The research has a certain degree of innovation, which provides a great reference for engineering practice and can be accepted for publication in journals.

Reviewer #2: (No Response)

7. PLOS authors have the option to publish the peer review history of their article (what does this mean?). If published, this will include your full peer review and any attached files.

Reviewer #1: No

Reviewer #2: No

---

## [Editor Report · Acceptance letter]

14 Jun 2021

PONE-D-21-11948R1 

Deterioration Characteristics of Cement-Improved Loess under Dry–wet and Freeze–thaw Cycles 

Dear Dr. Ni:

I'm pleased to inform you that your manuscript has been deemed suitable for publication in PLOS ONE. Congratulations! Your manuscript is now with our production department. 

Kind regards, 

on behalf of

Dr. Ahmed Mancy Mosa 

Academic Editor

PLOS ONE